# Family and Individual Contexts of Middle-School Years and Educational Achievement of Youths in Middle-Aged Adulthood

**DOI:** 10.3390/ijerph20043279

**Published:** 2023-02-13

**Authors:** Jerf W. K. Yeung, Lily L. L. Xia

**Affiliations:** 1Department of Social and Behavioral Sciences, City University of Hong Kong, Kowloon, Hong Kong, China; 2Department of Sociology, Zhejiang University, Hangzhou 310085, China

**Keywords:** family SES, parental support for college education, educational expectation, academic commitment, educational development

## Abstract

Although educational development of youths can profoundly affect their other domains of health and well-being across later life trajectories, little research has investigated the prolonged effects of family and individual contexts of youths in middle-school years, a most critical developmental and formative stage, on their educational achievement in middle-aged adulthood. The current study employed data of a nationwide representative sample of middle-school youth students in the Longitudinal Study of American Youth (LSAY) to examine how grade-7 parental support for college education, family SES, and educational expectations of youths contribute to their educational achievement in adulthood of mid-thirties through their development of grade-8 academic commitment and grade-9 educational performance in terms of English, mathematics, science, and social studies grade scores. Results based on structural equation modeling of longitudinal relationship found that grade-7 parental support for college education, family SES, and educational expectations of youths had significant and direct effects on youths’ higher educational achievement in adulthood, and youths’ grade-8 academic commitment and grade-9 educational performance significantly mediated the effects of grade-7 family SES, parental support for college education, and educational expectations of youths on their educational achievement in adulthood respectively and/or concurrently. Furthermore, interaction analysis supported the promotive but not buffering effects of grade-7 educational expectations of youths by family SES on their grade-9 educational performance and educational achievement in adulthood. Implications related to the important findings of the current study pertaining to educational development of youths are discussed.

## 1. Introduction

The educational development of youths is pivotal to their health and well-being in later life trajectories [1,2]. Undeniably, family and youths’ individual characteristics are of fundamental importance to shape youths’ educational success in later life. Apparently, family is the most intimate and influential socialization agent contributing to various aspects of youth development including their educational performance and success [3,4,5]. Moreover, recent theoretical and empirical work reports that although youth development is susceptible to the impacts of their proximal socialization environment such as family conditions, youths themselves may cultivate their own development by personal agency [6,7], in which youths who are of higher positive future expectation and motivation would perform better academically and behaviorally. Furthermore, although youth development is a progressive stage from early adolescence to young adulthood, early adolescence of the middle-school years is deemed as the most critical developmental and formative period poised youths to establish their cognitive, behavioral, and educational foundations for future positive development and accomplishment [8,9]. Nevertheless, little existing research has examined the prolonged effects of family and individual contexts of youths in middle-school years on their later educational achievement in adulthood of mid-thirties, which is an important life stage for career development. To fill this research gap, we intend to investigate how family socioeconomic status (family SES), parental support for college education, and educational expectations of youths in grade 7 contribute to their educational achievement in adulthood of mid-thirties through the development of youths’ academic commitment in grade 8 and educational performance in grade 9, stretching a life span of youth educational development from early adolescence to mid-adulthood across 20 years.

Specifically, family SES and parental support for college education are important family factors constituting a positive family context conducive to youths’ academic attitudes and behaviors as well as educational development [10,11]. In addition, as aforementioned, youths may have their personal initiative and motivation to plan and establish their own developmental path by personal agency [5,7], in which educational expectations of youths are promotive of youths’ educational development by facilitating their academic commitment and engagement [12]. In this study, we anticipate that family SES, parental support for college education, and educational expectations of youths in grade 7 would positively predict youths’ grade-8 academic commitment and their grade-9 educational performance, which then act together contributing to youths’ educational achievement in adulthood of mid-thirties. Accordingly, academic commitment and educational performance of youths are believed to mediate the effects of family SES, parental support for college education, and youths’ educational expectations on their educational achievement in adulthood. Furthermore, although family SES has been consistently confirmed as a powerful structural contributor of youths’ educational success, parental educational support and engagement and youths’ individual motivation and commitment are also considered as important makers of youths’ educational success in long run [12,13], especially for those living in lower family SES. Hence, in this study we also investigate how parental support for college education and youths’ educational expectations moderate the effects of family SES on their academic commitment and educational development in middle school and adulthood.

### 1.1. Family Context and Educational Development of Youths

Theoretically and empirically, family socialization provides the most intimate and influential processes to shape various aspects of youth cognitive, psychosocial, and educational development. This is consonant with the life course perspective stressing that family is the primary and pivotal socialization arena to cultivate perceptions, experiences, values, interpretations, and behaviors of youths toward external wider social world [5,14]. For educational development of youths, family SES and parental support for college education are thought as closely relevant socialization components at the family context relating to youths’ educational success [11,12]. Family SES is commonly referred to the education, income, and occupation possessed by parents [11], which is germane to the availability of resources, human and social capitals, learning stimulations, academic guidance, and educational opportunities for youths’ educational development. Empirical research consistently supported that family SES positively predicted educational performance and outcomes of children and youths [4,11,15]. Family stress theory explains that families of low socio-economic conditions may face more daily difficulties and challenges for parents to cultivate their children in a constructive and pro-education way due to lack of financial, informational, and social resources and supports [16,17]. Family scholars reported that parents of low family SES generally have insufficient human and social capitals, long working hours, unstable employment, limited incomes, and inadequate coping and parenting strategies [17,18], which heavily undermine parental engagement and planning for their children’s better educational development [18,19]. In a study by Chen, et al. [20], they found that family SES positively predicted reading ability of middle-school students. Recently, Zhang, et al. [9] confirmed that family SES positively predicted youths’ academic achievement in terms of Chinese and mathematics scores 9 months later while controlling for prior academic achievement. In an earlier meta-analysis conducted by White [21] on reviewing around 200 studies, he found a substantial correlation of r = 0.35 between SES and academic achievement. In a more recent meta-analysis by Sirin [19], he also reported the substantial association between SES and academic achievement with the average effect-size for the fixed effects, ES = 0.28, and the average effect-size for the random effects, ES = 0.27. Scholars hence claimed that the association between family SES and youths’ educational development is SES-related gaps in academic performance [22], indicating that family socioeconomic conditions can be a key structural factor at family context to differentiate youths’ educational development and success.

Moreover, it is believed that the relationship between family SES and youths’ educational development is indirect through youths’ academic attitude and behavior, in which higher family SES is conducive to youths’ development of better academic commitment that in turn may lead to their later better educational performance and achievement. This is valid as youths’ educational attitudes and behavior are explicitly shaped by tangible family academic socialization conditions, such as the availability of reference books, calculators, computer equipment, study area, and parental knowledge and guidance [9,17]. Relevantly, social capital theory accentuates that more affluent and resourceful families can take an advantage of cultivating their children with higher educational motivation and development by providing them better constructive home learning environment and conditions [23]. Rivas-Drake and Marchand [24] mentioned that “socioeconomic status (SES) is positively related to youths’ academic outcomes in part through involvement in their children’s academic socialization (p. 227).” Recently, Tomaszewski, et al. [25] found that family SES was positively related to educational engagement of a representative sample of Australian adolescent students that in turn mediated the effects of family SES on their academic achievement measured by standardized test scores. Pertinently, Berger and Archer [26] reported that adolescent students coming from a school of lower SES significantly showed less endorsement of educational motivation than their peers studying in a school of higher SES. For this, we anticipate that family SES would promote youths’ academic commitment and educational performance in middle school and educational achievement in adulthood, in which youths’ academic commitment and educational performance would mediate the effect of family SES on youths’ educational achievement in adulthood.

Moreover, although research confirmed that family SES is positively influential of parents’ engagement in educational development of their children, some recent research reported that parents may still hold educational concern and support for their youth children’s educational success even under disadvantaged family conditions [10,27]. The expectancy-value family socialization theory can help explain this seemingly paradoxical phenomenon of parental positive educational support [28], in which if parents reckon the practical utility and values of education, they are more willing to expect and support for better educational development of their children even in difficult family situations. Rivas-Drake and Marchand [24] found that even adjusted for parental education, a strong indicator of family SES, educational expectations of Latino parents were the only significant predictor of the utility of education among their adolescent children. Likewise, Sonnenschein, et al. [29] reported that low-SES Black and Latino Head Start parents’ beliefs and support for math and reading socialization for their children significantly predicted their children’s engagement in reading and math activities, which then led to their better vocabulary and early math skills. For this, Yamamoto and Sonnenschein [22] mentioned that “(a) growing body of evidence in the United States has demonstrated strengths and resilience shared by ethnic minority and immigrant families, including those living in poverty, and creative strategies they use to navigate their children’s education (p. 184).” Besides, as academic attitudes and educational behavior of youths are profoundly affected by their parental educational support, it is hence plausible to expect that parental support for college education would indirectly lead to educational achievement of youths in adulthood through affecting youths’ development of academic commitment and educational performance in middle school. According to the intergenerational transmission perspective, parents are the most influential intimate figures inheriting their offspring the values, beliefs, behavioral choices, and life directions through the family modeling processes [30]. Thereby, if youths receive more parental educational support, they may develop higher academic commitment and have better educational performance in middle school, which then contribute to their educational achievement in adulthood [28,31]. Accordingly, we anticipate that parental support for college education of their youth children would promote youths’ academic commitment and educational performance in middle school and educational achievement in adulthood, in which youths’ academic commitment and educational performance would mediate the effect of parental support for college education on youths’ educational achievement in adulthood.

### 1.2. Individual Context of Youths and Their Educational Development

Although educational development of youths is profoundly shaped by the socialization process of family context [4,32], as aforementioned youths themselves may individually and initiatively direct and plan their own life path and development by personal agency even facing environmental challenges [15,33]. This proposition is insightful to help explaining why some youths can achieve better educational success even growing up in unfavorable environmental socialization conditions, revealing the proactive functioning of youths in interplay with environmental influences on their educational development [13,33]. The self-determination theory states that although individuals are susceptible to the impacts of their living environments, they may have their own wills and tendency to develop their life path and direction [34], propelling their efforts and motivation for achieving the set goals. Pertinently, by using data from the Program for International Student Assessment 2018 (PISA), Ali, et al. [35] found that the first- and second-generation immigrant youths significantly had higher educational expectations than their native peers even controlling for economic, social, and cultural status, which in turn predicted their higher reading achievement. More relevantly, Renzulli and Barr [36] reported that both the degree and institutional route expectations of grade-9 adolescent students significantly predicted their degree expectation in grade 11 while accounting for family SES and family economic shocks, in which the degree and institutional route expectations of adolescent students were less adversely affected by family SES and family economic shocks in the lower socioeconomic-status group compared to their counterparts in middle-class status. As educational expectations of youths have been empirically reported as valuable individual resources promotive of youths’ educational development [37,38], we anticipate in this study that educational expectations of youths would promote youths’ academic commitment and educational performance in middle school and educational achievement in adulthood, in which youths’ academic commitment and educational performance would mediate the effect of youths’ educational expectations on their educational achievement in adulthood.

Moreover, the expectancy value theory assumes that one’s engagement in carrying out certain behaviors or actions for achieving a specific desired outcome or goal depends on her/his expectancy, desire, and values placed on the outcome or goal set [12]. If a person seriously regards and expects for achieving a certain outcome or goal, i.e., achieving higher education, she/he will be more actively committed and engaged to perform the corresponding behaviors and actions for attaining the set outcome or goal regardless the obstacles of external environmental challenges, e.g., lower family SES. In their recent study, Chen, et al. [20] found that the positive effect of family SES on reading ability of Chinese youths was significantly moderated by youths’ learning motivation, in which the direct effects of family SES on their reading ability at high, medium, and low levels of learning motivation were β= 0.24, 0.32, and 0.40, respectively, connoting the abating effect of family SES on the reading ability of Chinese youths with higher learning motivation. As educational expectations of youths are future-oriented in nature that imply its persistence in interacting with environmental impacts, we hence anticipate that educational expectations of youths would moderate the effects of family SES on their academic commitment and educational performance in middle school and educational achievement in adulthood, in which the effects of family SES on youths’ academic commitment and educational development would be less pronounced among youths of higher educational expectations.

### 1.3. The Current Study 

In this study, we attempt to examine how family SES, parental support for college education, and educational expectations of youths in grade 7 predict youths’ grade-8 academic commitment and grade-9 educational performance in terms of English, mathematics, science, and social studies grade scores and their latent composite scores, which are considered importantly contributive to the educational achievement of youths in adulthood. Specifically, family SES, parental support for college education, and educational expectations of youths in grade 7 are anticipated to predict grade-8 academic commitment and grade-9 educational performance of youths and their educational achievement in adulthood directly and concurrently. Besides, grade-8 academic commitment of youths is anticipated to mediate the effects of grade-7 family SES, parental support for college education, and educational expectations of youths on their grade-9 educational performance and educational achievement in adulthood. In addition, grade-8 academic commitment of youths and their grade-9 educational performance are anticipated to respectively and concertedly mediate the effects of grade-7 family SES, parental support for college education, and educational expectations of youths on educational achievement of youths in adulthood. Furthermore, parental support for college education and educational expectations of youths in grade 7 are anticipated to moderate the effects of grade-7 family SES on youths’ academic commitment in grade 8 and educational performance in grade 9 as well as educational achievement in adulthood. Taken together, we have the following hypotheses:

**H1.** 
*It is hypothesized that family SES, parental support for college education, and educational expectations of youths in grade 7 would positively predict youths’ academic commitment in grade 8 and educational performance in grade 9 and educational achievement in adulthood.*


**H2.** 
*It is hypothesized that academic commitment of youths in grade 8 would positively predict youths’ educational performance in grade 9 and educational achievement in adulthood, and would mediate the effects of family SES, parental support for college education, and educational expectations of youths in grade 7 on youths’ educational performance in grade 9 and educational achievement in adulthood.*


**H3.** 
*It is hypothesized that educational performance of youths in grade 9 would positively predict youths’ educational achievement in adulthood, and would mediate the effects of family SES, parental support for college education, and educational expectations of youths in grade 7 on youths’ educational achievement in adulthood.*


**H4.** 
*It is hypothesized that academic commitment of youths in grade 8 and their educational performance in grade 9 would in alliance mediate the effects of family SES, parental support for college education, and educational expectations of youths in grade 7 on educational achievement of youths in adulthood.*


**H5.** 
*It is hypothesized that parental support for college education and educational expectations of youths in grade 7 would moderate the effects of grade-7 family SES on youths’ academic commitment in grade 8 and educational performance in grade 9 and educational achievement in adulthood to a lesser degree.*


For precluding the artifacts of sociodemographic confounders, gender, family structure, and ethnic origin of youths are controlled as covariates. This is important as female youth students are reported more academically committed and have higher educational attainments [1,39], and youths from two-parent family of both biological mother and father demonstrate better learning attitudes and behavior as well as academic performance [1,40]. In addition, youths of different ethnic origins may have different educational motivation and performance, in which Asian youths are reported to have more academic persistence and educational success compared to youths of other ethnic origins like Hispanics, Africans, and native/other Americans, and non-Hispanic Whites are in-between [1,41]. In this study, four ethnic dummy variables are constructed to represent youths of Hispanic, African, other, and non-Hispanic White origins with Asian youths as reference.

## 2. Methods

This study used data drawn from the Longitudinal Study of American Youth (LSAY), a nationwide study conducted in the United States with middle- and high-school students, parents, and teachers in public schools [42]. LSAY was funded by the National Science Foundation (NSF) to follow two cohorts of students starting in 1987. The first cohort contained 2829 high-school students in grade 10 and the second cohort included 3116 middle-school students in grade 7. The longitudinal surveys were administered annually for 7 years ending in 1994, with 3 years of high school and 4 years after high school for cohort-1 students, and with 3 years of middle school and 3 years of high school and 1 year after high school for cohort-2 students. In 2006 NSF provided additional funding to LSAY for tracking educational and occupational development of the original sample of cohort-1 and cohort-2 students, in which five additional longitudinal surveys were conducted in 2007, 2008, 2009, 2010, and 2011. In 2007, LSAY successfully relocated around 95% of the original sample of 5945 students in both cohorts, during which the student participants were aged 33 to 37 years old. The sampling method of LSAY was based on a stratified sampling framework from a national population of middle and high schools in 12 sampling strata defined by the geographic region and type of community of the country, in which about 60 grade-7 and grade-10 students in each sampled school were randomly selected respectively. Hence, the sample of LSAY was representative of the middle- and high-school students in public schools in the country. LSAY aims to trace educational and STEM as well as occupational development of American youth students by surveying their family and sociodemographic background, learning attitudes and behavior, educational experiences and achievement, out of school activities, post-high school plans, psychosocial characteristics, and educational support from parents, teachers, and peers, among other variables. The original sample approximately comprised of 48% female and 52% male students, and 70% European Americans, 17% African Americans, 9% Hispanics, 3% Asian Americans, and 1% Native and other Americans. The present study adopts cohort-2 students of LSAY as the study sample due to the data of cohort-2 students containing family and individual characteristics and youths’ educational development across middle-school years to adulthood.

### 2.1. Measures

**Family SES** was measured by the Duncan’s Socioeconomic Index (SEI) score when the youths were in grade 7, which was provided by the principal investigators of LSAY by calculating the parental responses of their educational level, income, and occupational prestige [43]. SEI is constructed by weighting an occupation’s median education and income on the metric of occupational prestige [43], which has been widely used to indicate family SES in empirical research with higher scores representing better family SES [44,45].

**Parental support for college education** was measured in grade 7 by an item of “parent college push grade 7” provided by the principal investigators of LSAY, in which the item was rated in a 6-point scale ranging from 0 = low parent college push to 5 = high parent college push, meaning that higher scores indicate more parental college push. Concretely, this measure has been commonly used to tap on parents’ educational support for their children to attain college education [46]. 

**Educational expectations of youths** were measured in grade 7 by an item “What is the highest level of education that you think you will have completed by the time you are 40?”, which was responded by a 6-point scale of 1 = high school only, 2 = vocational/trade school, 3 = some college, 4 = bachelor’s degree, 5 = master’s degree, and 6 = doctorate/professional degree. Hence, higher scores represent more positive educational expectations of the youths [47], which are important for youths’ educational development and success. 

**Academic commitment of youths** was measured in grade 8 by two items, which include “Try hard to do my best in school” and “Try harder if I get bad grades” with an introduction sentence “How often do you do each of the following?”. The items have been commonly used to indicate youths’ academic commitment and motivation [48], and were rated by a 5-point scale ranging from 1 = never to 5 = always. The two items are combined to form a composite score, in which higher scores indicate more youths’ academic commitment. Cronbach’s alpha of the measure is adequate, α = 0.684.

**Educational performance of youths** in grade 9 was measured by their grade scores in the subjects of English, mathematics, science, and social studies rated by a 8-point scale of 1 = mostly A, 2 = half A and half B, 3 = mostly B, 4 = half B and half C, 5 = mostly C, 6 = half C and half D, 7 = mostly D, and 8 = mostly below D. The items were inversely coded and therefore higher scores connote better educational performance of youths in English, mathematics, science, and social studies. In fact, the grade scores of youths’ English, mathematics, science, and social studies are highly correlated with each other, r = 0.411 to 0.574, *p* < 0.001, and the composite reliability of the grade scores examined by confirmatory factor analysis with robust maximum likelihood (MLR) estimation is excellent *ρc* = 0.797, with model fit of comparative fit index (CFI) = 0.993, root mean square error of approximation (RMSEA) = 0.052, and standardized root mean-square residual (SRMR) = 0.015, supporting the convergence of the grade scores of English, mathematics, science, and social studies on a latent construct of educational performance of youths in later analysis [49]. 

**Educational achievement of youths** in adulthood of mid-thirties was measured by an item of “Highest level of education of the youth participant in 2007” rated by six categories, 1 = Less than high school diploma, 2 = High school diploma, 3 = Associates degree, 4 = Baccalaureate degree, 5 = Master’s degree, and 6 = Doctorate/Professional degree. Hence, higher scores mean better educational achievement attained by the youths in adulthood.

**Covariates** of youths’ gender, family composition, and ethnicity were controlled in the modeling procedures to preclude confounding effects. Youths’ gender (1 = female, 2 = male) and family composition (1 = otherwise, 2 = two-parent family) are dichotomous variables, and youths’ ethnicity is classified into four dummy variables with youths of Hispanic, African, non-Hispanic White, and other origins = 1 and Asian youths = 0.

### 2.2. Modeling Procedures

Structural equation modeling (SEM) of longitudinal relationship is used to model the effects of grade-7 family SES, parental support for college education, and educational expectations of youths on educational achievement of youths in adulthood through the mediation of youths’ grade-8 academic commitment and grade-9 educational performance. The basic form of a SEM model contains two parts: the structural model and measurement models, which have
(1)η=Bη+Γξ+ζ
(2)$$Y=\Lambda_{y}\eta +\varepsilon$$
(3)$$X=\Lambda_{x}\xi +\delta .$$

The first equation is the structural model to construct the study relationships among the study variables, in which *η* represents endogenous variables and *ξ* refers to exogenous variables that are connected with a system of linear equations by the coefficient matrices of beta *B* and gamma *Γ*, as well as zeta *ζ* as residual vector. Specifically, *Γ* represents effects of exogenous variables on endogenous variables, which refer to grade-7 family SES, parental support for college education, and educational expectations of youths in prediction of youths’ grade-8 academic commitment, grade-9 educational performance, and educational achievement in adulthood; and *B* represents effects of endogenous variables on other endogenous variables that in our study are youths’ grade-8 academic commitment and grade-9 educational performance in relation to their educational achievement in adulthood, and *ζ* indicates the regression residual terms. 

Equations (2) and (3) are the measurement models to define latent variables by the observed variables or manifest variables. Equation (2) regresses the observed *y* variables on endogenous latent variables or referred as endogenous manifest variables by its single indicators *η*, which mean that the indicators of English, mathematics, science, and social studies grade scores are loaded on the latent construct of educational performance of youths in grade 9 or are simply indicated as individual manifest variables, and academic commitment of youths and their educational achievement in adulthood are measured as manifest variables. Equation (3) regresses the observed *x* variables on the exogenous latent variables or directly measured as manifest exogenous variables by its single indicator *ξ*, meaning that the single indicators of grade-7 family SES, parental support for college education, and educational expectations of youths are measured as respective exogenous manifest predictors. For this, *Λy* refers to the endogenous variables of youths’ academic commitment in grade 8, educational performance in grade 9, and educational achievement in adulthood, and *Λx* means the predicator variables of grade-7 family SES, parental support for college education, and educational expectations of youths, and *ε* and *δ* are the measurement errors. 

Modeling procedures were fit in Mplus 8.6 [50]. Due to data of LSAY nested in school level and our study concerning family and individual effects of youths on educational development of youths from middle-school years to adulthood, the COMPLEX function was used with CLUSTER by setting <type = complex> to account for interdependence of the data structure by adjusting standard errors and chi-square tests [51]. Full information maximum likelihood (FIML) method was employed to allow data of all cases being used for model estimations [52], which can generalize results to the population in longitudinal studies. Model fit was evaluated by comparative fit index (CFI), root mean-square error of approximation (RMSEA), and standardized root mean-square residual (SRMR). The threshold for acceptable model fit is: CFI > 0.90, RMSEA < 0.08, and SRMR < 0.1; and excellent model fit is: CFI > 0.95, RMSEA< 0.06, and SRMR < 0.08 [51].

## 3. Results

In our sample, female and male students shared 48% (*n* = 1495) and 52% (*n* = 1621) respectively, and most of them came from two-parent families (87.1%, *n* = 2715). For ethnicity, non-Hispanic Whites were the majority, (69.6%, *n* = 2169), followed by African Americans (16.2%, *n* = 504), Hispanics (9.1%, *n* = 284), Asians (3.5%, *n* = 112), and youths of other ethnic origins (1.5%, *n* = 47). Accordingly, the sociodemographic characteristics of the cohort-2 study sample in term of gender and ethnic composition are similar to the original sample reported in LSAY. For correlations of the study variables, Table 1 shows that the predictors of grade-7 family SES, parental support for college education, and educational expectations of youths were significantly and substantially correlated with each other, r = 0.246 to 0.365, *p* < 0.001, which were also significantly and positively correlated with youths’ grade-8 educational commitment, r = 0.078 to 0.250, *p* < 0.001, grade-9 educational performance of English, mathematics, science, and social studies grade scores, r = 0.124 to 0.309, *p* < 0.001, as well as educational achievement in adulthood, r = 0.329 to 0.485, *p* < 0.001. In addition, grade-8 academic commitment of youths was significantly correlated with youths’ grade-9 educational performance of English, mathematics, science, and social studies grade scores, r = 0.183 to 0.234, *p* < 0.001, and educational achievement in adulthood, r = 0.226, *p* < 0.001. Besides, grade-9 educational performance of youths was significantly correlated with their higher educational achievement in adulthood, r = 0.291 to 0.400, *p* < 0.001. 

The first structural equation model of longitudinal relationship was conducted to examine the effects of grade-7 family SES, parental support for college education, and educational expectations of youths on youths’ grade-9 educational performance of English, mathematics, science, and social studies grade scores through the mediation of youths’ grade-8 academic commitment while adjusting for youths’ gender, family structure, and ethnic origins (model 1). An excellent model-data fit was obtained: CFI = 1.00, RMSEA = 0.00, and SRMR = 0.00. Figure 1 presents the standardized results, in which grade-7 educational expectations of youths had the strongest significant and positive effects on youths’ grade-9 English, mathematics, science, and social studies grade scores, β = 0.174, 0.148, 0.204, and 0.209, *p* < 0.001. Moreover, grade-7 parental support for college education and family SES also significantly predicted youths’ better grade-9 educational performance in terms of English grade score, β = 0.076 and 0.065, *p* < 0.001 and 0.01; mathematics grade score, β = 0.059 and 0.051, *p* < 0.01 and 0.05; science grade score, β = 0.088 and 0.103, *p* < 001; and social studies grade score, β = 0.078 and 0.076, *p* < 0.001. Besides, parental support for college education and educational expectations of youths significantly predicted higher grade-8 academic commitment of youths that in turn significantly contributed to youths’ better grade-9 English, mathematics, science, and social studies grade scores, β = 0.165, 0.136, 0.153, and 0.166, *p* < 0.001. In addition, the outcomes of youths’ grade-9 English, mathematics, science, and social studies grade scores were significantly correlated with each other substantially and positively, in which the standardized covariances were between r = 0.338 to 0.496, *p* < 0.001, corroborating valid to converge them on a latent construct of grade-9 educational performance of youths.

To examine whether grade-8 academic commitment of youths mediated the effects of grade-7 family SES, parental support for college education, and educational expectations of youths on their grade-9 educational performance of English, mathematics, science, and social studies grade scores, indirect-effect tests were conducted. Table 2 shows that grade-8 academic commitment of youths significantly mediated the effects of grade-7 parental support for college education on youths’ grade-9 English, mathematics, science, and social studies grade scores concurrently, β_ind_ = 0.015, 0.012, 0.014, and 0.016, *p* < 0.01. In addition, grade-8 academic commitment of youths also significantly mediated the effects of grade-7 educational expectations of youths on their grade-9 English, mathematics, science, and social studies grade scores concurrently, β_ind_ = 0.045, 0.038, 0.043, and 0.048, *p* < 0.001. However, grade-8 academic commitment of youths did not have any mediated effects on the relationship between grade-7 family SES and youths’ grade-9 educational performance of English, mathematics, science, and social studies grade scores. This is valid as family SES was not significantly related to youths’ academic commitment in grade 8. For the moderated effects of grade-7 parental support for college education and educational expectations of youths by grade-7 family SES on youths’ grade-8 academic commitment and their grade-9 educational performance, interaction analysis found that only educational expectations of youths significantly interacted with family SES in prediction of youths’ grade-9 English, mathematics, and science grade scores, which are in a promotive but not buffering way (Table 3), β_int_ = 0.052, 0.049, and 0.048 *p* < 0.01 and 0.05.

The second structural equation model of longitudinal relationship was conducted to test the effects of grade-7 parental support for college education, family SES, and educational expectations of youths on youths’ educational achievement in adulthood through the mediators of youths’ grade-8 academic commitment and grade-9 educational performance while adjusting for youths’ gender, family structure, and ethnic origins (model 2). The model-data fit was excellent: CFI = 0.993, RMSEA = 0.019, and SRMR = 0.012, in which the English, mathematics, science, and social studies grade scores were all significantly and well adequately loaded on the latent construct of grade-9 educational performance of youths in model 2, λ = 0.711, 0.587, 0.753, and 0.756, *p* < 0.001. Figure 2 shows that grade-7 parental support for college education, family SES, and educational expectations of youths significantly and positively predicted educational achievement of youths in adulthood respectively, β = 0.095, 0.105, and 0.253, *p* < 0.001. Moreover, grade-7 parental support for college education and educational expectations of youths also significantly predicted youths’ higher grade-8 academic commitment, β = 0.056 and 0.224, *p* < 0.01 and 0.001, and grade-9 educational performance, β = 0.108 and 264, *p* < 0.001, which in turn significantly contributed to youths’ higher educational achievement in adulthood, β = 0.040 and 0.327, *p* < 0.05 and 0.001. Nevertheless, grade-7 family SES only significantly predicted higher educational performance of youths in grade 9 and educational achievement in adulthood, β = 0.108 and 0.015, *p* < 0.001, but not their grade-8 academic commitment, β = −0.001, *p* > 0.05. 

Followed by indirect-effect tests were conducted to examine whether grade-8 academic commitment of youths and/or their grade-9 educational performance mediated the effects of grade-7 parental support for college education, family SES, and educational expectations of youths on educational achievement of youths in adulthood. Table 4 shows that the effect of grade-7 parental support for college education on educational achievement of youths in adulthood was significantly mediated by youths’ grade-8 academic commitment, β_ind_ = 0.003, *p* < 0.05, and their grade-9 educational performance respectively, β_ind_ = 0.041, *p* < 0.001, as well as the combination of youths’ grade-8 academic commitment and grade-9 educational performance, β_ind_ = 0.005, *p* < 0.01. Moreover, the effect of youths’ grade-7 educational expectations on their educational achievement in adulthood was significantly mediated by their grade-8 academic commitment, β_ind_ = 0.008, *p* < 0.05, and grade-9 educational performance respectively, β_ind_ = 0.076, *p* < 0.001, as well as the combination of youths’ grade-8 academic commitment and their grade-9 educational performance, β_ind_ = 0.014, *p* < 0.01. However, the effect of grade-7 family SES on educational achievement of youths in adulthood was only significantly mediated by youths’ grade-9 educational performance, β_ind_ = 0.003, *p* < 0.001. For the interaction effects of grade-7 parental support for college education and educational expectations of youths by family SES on youths’ grade-8 academic commitment, grade-9 educational performance, and educational achievement in adulthood, interaction analysis found that grade-7 parental support for college education significantly interacted with family SES in prediction of youths’ better grade-9 educational performance (Table 5), β_int_ = 0.002, *p* < 0.05. Besides, grade-7 educational expectations of youths significantly interacted with family SES in prediction of youths’ higher grade-9 educational performance, β_int_ = 0.003, *p* < 0.01, and also their educational achievement in adulthood, β_int_ = 0.002, *p* < 0.01. Nevertheless, the significant moderation effects are all in a promotive but not buffering way.

## 4. Discussion 

With the use of four-wave longitudinal data across 20 years from a nationwide representative youth sample of LSAY, this study investigated the prolonged positive effects of grade-7 parental support for college education, family SES, and youths’ own educational expectations on youths’ educational achievement in adulthood of mid-thirties through the mediation of youths’ grade-8 academic commitment and grade-9 educational performance. The results added empirical support to the importance of family socialization and individual agency of youths on their academic commitment and educational performance and achievement from early adolescence to adulthood [12,53]. The prolonged effects of familial and individual contexts of youths in middle-school years on their educational achievement in adulthood were substantially supported in the current study, which can contribute to the existing literature by shedding light on the rudimental influences of early adolescence in relation to youths’ immediate and long-term educational development [1,54]. Adolescence is a period of life transformation where young people build up cognitive, psychosocial, and educational bricks for their future achievement and all-round well-being [8,55]. The educational attitudes, behaviors, and performance of youths during this period are usually referred as critical indicators reflecting the status of individual development in adolescence, which are pivotally influential of their future educational achievement and success [1,9]. Educational development of youths has long been reckoned as one of the most fundamental human capitals and assets for young people to break through the intergenerational heritance of social and family disadvantages for better upward mobility [11,56]. Therefore, revelation of the promoting factors and mechanisms of youths’ education-related qualities would provide guidance for pertinent policy making and service planning. 

In this study, parental support for college education, family SES, and educational expectations of youths in grade 7 were found as important and independent contributors in relation to youths’ educational commitment in grade 8, educational performance in grade 9, and educational achievement in adulthood concurrently and differently. Specifically, the findings of the current study provide fruitful information for policy design, educational innovation, and allocation of resources to target parental engagement and support for their children’s education, family financial and resource status of youths, and youths’ development of educational expectations. Pertinently, parental support for college education was found significantly predictive of youths’ academic commitment, educational performance, and educational achievement over time started from grade 7, across middle-school years, and stretching to adulthood, which evinces that the impacts of parental educational engagement and support on youths’ educational development and success are persistent and independent of the influences of family SES and youths’ own educational expectations. This supports the life course perspective positing that early experiences of children in home with their parental figures can become solid cognitive and behavioral foundations influential of their later educational and social development [5,14]. Relevantly, it is also consistent with the intergenerational transmission perspective that parents pass on their offspring family advantages or disadvantages by parental expectations and socialization, such as parental support for college education in the current study, to shape the later educational trajectories of their children [15]. In fact, family researchers agreed that parental figures are the most intimate and influential socialization agent influential on youths’ education and other aspects of development [33,57]. 

Moreover, the importance of family SES in youths’ educational development and success was also confirmed in the current study, which aligns the results of previous research [18], in which except its insignificant effect on youths’ grade-8 academic commitment, family SES significantly predicted youths’ grade-9 educational performance and educational achievement in adulthood. The findings are congruent with the social capital theory that educational development of youths is susceptible to the tangible learning and stimulating environments provided by their family [4,15,28], in which parents of more economic, cultural, and social resources can better cater for the educational needs and success of their children. However, supported by the family stress theory that lower family SES denotes more family difficulties and challenges, which are directly inimical to educational development of youths [16,17]. As family SES tells the availability of parental human capitals, family resources, home stimulations, and learning environment available for youth development [18,19], policy makers, educators, and youth practitioners should pay more attention and assistance to those youths from economically disadvantaged families. Interestingly, educational expectations of youths were found significantly and substantially contributive to youths’ grade-8 academic commitment, grade-9 educational performance, and educational achievement in adulthood positively, in which the effects are strongest compared to parental support for college education and family SES. This explicates the importance of youths’ personal agency for their positive development [1,7,12]. This corresponds to the self-determination theory and expectancy value theory that youths themselves can act as an active agent to self-determine and plan their educational directions and development even at the same time they are affected by the contexts of parental socialization and family tangible conditions [7,12], in which if youths conceive higher educational expectations, they can perform better in their educational development. Hence, helping youths develop and maintain positive personal agency should be considered as crucial policy and educational concerns. Accordingly, the direction of policy and service formulations should be multi-faceted and integrated systematically concerning both parental educational engagement for their children, family financial and tangible needs, and youths’ development of educational expectations, that is, the policies and services should work on the parent side to promote their positive attitude towards children’s education and parental ability of cultivating their youth children’s educational attitude and behaviors, and/or provide tangible learning supports for youths living in economically disadvantaged families, and/or implement at youths’ individual level to promote youths’ educational expectations and improve their learning skills and academic performance, and/or conduct the above-mentioned policy and service interventions concomitantly at different levels by parent-child joint programs. 

Undeniably, adolescence is a formative and developmental stage, in which educational development of youths is manifestly susceptible to the influences of multiple sources [2], such as family, school, peers, and youths’ own intrapersonal changes, with the most immediate and influential role played by the family. However, results of the current study revealed that youths themselves may play an even more active and prominent role instead of a passive one in the process of educational development. Consistent with the findings of previous research, youth themselves can achieve higher educational and social success if they can hold a strong motivation and commitment for their own positive development even though in disadvantaged conditions such as low family SES or parents’ passive attitudes towards education [33,35]. This is supportive of the thesis of youths’ personal agency in relation to their development and life trajectories [6,15], which is evidenced here to powerfully impact youths’ middle-school educational performance and educational achievement in mid-thirties directly and indirectly in a far-reaching way. In fact, the mediation analysis of this study further supports the importance of youths’ family and personal factors in early adolescence in relation to youths’ educational development indirectly and expansively. Specifically, youths’ grade-8 academic commitment significantly mediated the effects of parental support for college education and youths’ educational expectations in grade 7 on their grade-9 English, mathematics, science, and social studies grade scores respectively. Besides, youths’ grade-8 academic commitment and grade-9 latent educational performance by loading English, mathematics, science, and social studies grade scores convergently also significantly mediated the effects of grade-7 parental support for college education and youths’ educational expectations on youths’ educational achievement in adulthood. For this, it is plausible that attitudinal and behavioral educational support of parents and youths’ personal educational motivation are more important contributors than family SES to indirectly affect youths’ educational development, in which the latter may only indicate the environmental dimension of learning resources and stimulations but not the cultivation of youths’ intrapersonal positivity toward educational development. Therefore, strength-based and resilience-oriented policies and social services should be planned and implemented to support parents’ engagement in educational development of their youth children and also need to promote youths’ positive self-regard and their personal agency. In addition, due to the direct effects of family SES on youths’ grade-9 educational performance and educational achievement in adulthood significantly found, policy and program supports for youths living in disadvantaged families are important for promoting youths’ educational development.

On the other hand, interaction analysis of the current study did find some moderation effects of youths’ educational expectations by family SES and parental support for college education by family SES in grade 7 on youths’ educational performance in grade 9 and educational achievement in adulthood in a promotive way, which are contradictory to our hypothesis. Explicably, the significant and positive moderation effects of youths’ educational expectations by family SES on youths’ grade-9 English, mathematics, and science grade scores and educational performance in general as well as educational achievement in adulthood reveal that youths of higher educational expectations and living in better socioeconomic-status families have more prominent educational development than their counterparts either of higher educational expectations or having better family SES. Besides, the moderation effect of parenting support for college education by family SES was significantly and positively predictive of youths’ grade-9 educational performance, indicating that parental support for college education is more conducive to youths’ educational development in middle school under the condition of better family SES. These findings indicate that youths’ educational development is both benefited from better family SES and higher youths’ educational expectations and/or parental support for college education, indicating that socioeconomic situations and parental and youths’ individual factors that are all critical for youths’ educational development overarchingly [13,18]. For better helping youths plan and establish educational success, all these factors should be taken into account at policy, educational, family, and individual levels. 

## 5. Conclusions

In conclusion, this study has concurrently examined the influences of parental support for college education, family socioeconomic conditions, and educational expectations of youths on youths’ academic commitment and educational performance in middle-school years and their educational achievement in adulthood, which cover a twenty-year life span of youth educational development. The important findings of this study revealed the interrelated mechanism and mutual impacts of the familial and individual contexts on youths’ educational achievement in adulthood through the interrelated academic paths of youths’ academic commitment and educational performance during middle school by the use of longitudinal data. This evinces the evolving and accumulating nature of youths’ educational development at the family and individual processes, which have referential values for policy, educational, and youth service innovations. These include the needs of helping youths establish an integrative supportive learning environment at family and individual domains. Nevertheless, the current study contains some limitations that should be improved in the future. First, this study only examined family and youths’ individual factors in relation to youths’ educational development in middle school and adulthood. However, educational development of youths is influential by multiple systems at school, family, peers, individual, and community levels coincidently, which should be examined together in future research. Second, although middle-school years of youths are the crucial formative period affecting youths’ educational development, adolescence is a developmental and changing process that should also consider the stage of youths’ high-school academic performance and changes in relation to their educational success in adulthood. Third, the current study only examined the educational development of youths in the United States, where is a highly developed and Anglophone country. Future research should use representative samples of youths in other countries of different cultural, sociopolitical, and economic backgrounds, by which researchers can more clearly comprehend educational development of youths systematically and comparatively at a global context. Fourth, although the data of LSAY used in the current study contain a representative sample of youths following a time span of 20 years from 1987 to 2007, more updated data of educational development of youths are needed as family and school policies and environments have been changed saliently in recent years. Hence, the findings of the current study may not accurately reflect the educational development of youths nowadays, and more timely research is needed. Lastly, the current study considered parental support for college education, family SES, and educational expectations of youths, and their academic commitment and educational performance as fixed variables measured at one point of time, which should be better treated as time-varying variables to examine their changing effects on educational development of youths. Therefore, researchers in the future need to employ a latent state-trait perspective with advanced modeling procedures to investigate the transitional process of youths’ educational development and success.

## Figures and Tables

**Figure 1 ijerph-20-03279-f001:**
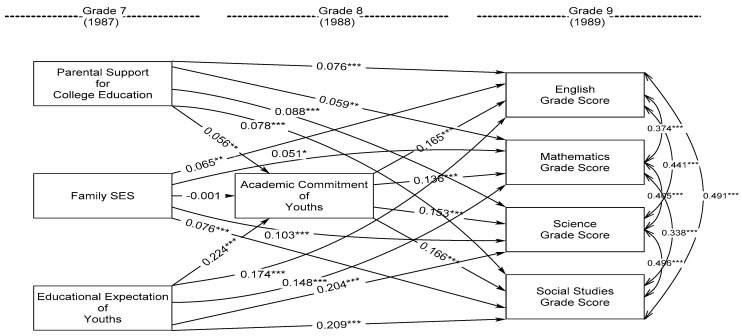
Standardized Effects of Parental Support for College Education, Family SES, and Educational Expectation of Youths in Grade 7 on Youths’ Grade-9 Educational Performance through Youths’ Grade-8 Academic Commitment (Model 1). Note. The gender, family structure, and ethnic origins of youths were adjusted in modelling procedures but skipped in presentation for simplicity. Family SES= family Socioeconomic Status. Model fit: CFI = 1.00, RMSEA = 0.00, and SRMR = 0.00. * *p* <0.05; ** *p* < 0.01; *** *p* < 0.001.

**Figure 2 ijerph-20-03279-f002:**
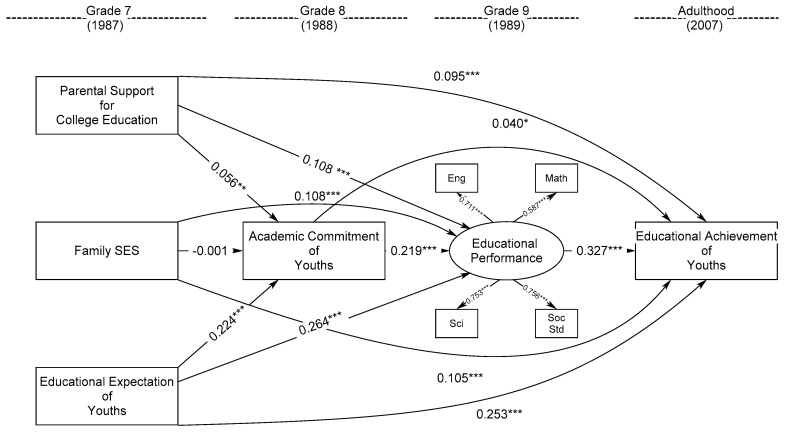
Standardized Effects of Parental Support for College Education, Family SES, and Educational Expectations of Youths in Grade 7 on Educational Achievement of Youths in Adulthood through Youths’ Grade-8 Academic Commitment and Grade-9 Educational Performance (Model 2). Note. The gender, family structure, and ethnic origins of youths were adjusted in modelling procedures but skipped in presentation for simplicity. Family SES = Family Socioeconomic Status. Eng = English Grade Score, Math = Mathematics Grade Score, Sci = Science Grade, Soc Std = Social Studies Grade Score. Model fit: CFI = 0.993, RMSEA= 0.019, and SRMR = 0.012. * *p* < 0.05; ** *p* < 0.01; *** *p* < 0.001.

**Table 1 ijerph-20-03279-t001:** Correlations of the Study Variables.

		1	2	3	4	5	6	7	8	9
1	Grade-7 Family SES	--								
2	Grade-7 Parental Support for College Education	0.246 ***	--							
3	Grade-7 Educational Expectations of Youths	0.290 ***	0.365 ***	--						
4	Grade-8 Academic Commitment of Youths	0.078 ***	0.136 ***	0.250 ***	--					
5	Grade-9 English Grade Score	0.162 ***	0.182 ***	0.273 ***	0.224 ***	--				
6	Grade-9 Mathematics Grade Score	0.124 ***	0.148 ***	0.221 ***	0.183 ***	0.442 ***	--			
7	Grade-9 Science Grade Score	0.198 ***	0.217 ***	0.312 ***	0.225 ***	0.514 ***	0.468 ***	--		
8	Grade-9 Social Studies Grade Score	0.174 ***	0.204 ***	0.309 ***	0.234 ***	0.557 ***	0.411 ***	0.574 ***	--	
9	Educational Achievement of Youths in Adulthood	0.355 ***	0.329 ***	0.485 ***	0.226 ***	0.359 ***	0.291 ***	0.400 ***	0.394 ***	--

Note. Family SES = Family Socioeconomic Status, *** *p* < 0.001.

**Table 2 ijerph-20-03279-t002:** Mediated effects of Grade-8 Academic Commitment of Youths on the Relationships of Grade-7 Parental Support for College Education, Family SES, and Educational Expectations of Youths in Prediction of Grade-9 Educational Performance of Youths in Model 1.

Predictor	Mediator	Outcome	β_ind_	SE	Z-Value
Parental Support for College Education	Academic Commitment of Youths	English Grade Score	0.015	0.005	3.122 **
		Mathematics Grade Score	0.012	0.004	2.910 **
		Science Grade Score	0.014	0.005	2.768 **
		Social Studies Grade Score	0.016	0.006	2.721 **
Family SES	Academic Commitment of Youths	English Grade Score	0.000	−0.028	0.978
		Mathematics Grade Score	0.000	−0.028	0.978
		Science Grade Score	0.000	−0.028	0.978
		Social Studies Grade Score	0.000	−0.028	0.978
Educational Expectation of Youths	Academic Commitment of Youths	English Grade Score	0.045	0.008	5.881 ***
		Mathematics Grade Score	0.038	0.008	4.763 ***
		Science Grade Score	0.043	0.008	5.604 ***
		Social Studies Grade Score	0.048	0.007	6.654 ***

Note. Family SES = Family Socioeconomic Status, ** *p* < 0.01; *** *p* < 0.001.

**Table 3 ijerph-20-03279-t003:** Moderated Effects of Parental Support for College Education and Educational Expectations of Youths by Family SES in Grade 7 on Youths’ Grade-8 Academic Commitment and Grade-9 Educational Performance (Model 1).

Predictor	Outcome	β_int_	SE	Z-Value
Parental Support for College Education X Family SES	Academic commitment of Youths	−0.001	0.014	−0.090
	English Grade Score	0.032	0.016	1.954
	Mathematics Grade Score	0.032	0.017	1.880
	Science Grade Score	0.012	0.015	0.811
	Social Studies Grade Score	0.022	0.014	1.602
Educational Expectations of Youths X Family SES	Academic commitment of Youths	0.007	0.019	0.358
	English Grade Score	0.052	0.018	2.824 **
	Mathematics Grade Score	0.049	0.019	2.603 **
	Science Grade Score	0.048	0.020	2.399 *
	Social Studies Grade Score	0.034	0.018	1.889

Note. Family SES = Family Socioeconomic Status, * *p* <0.05; ** *p* < 0.01.

**Table 4 ijerph-20-03279-t004:** Mediated Effects of Youths’ Grade-8 Academic Commitment and Grade-9 Educational Performance on the Relationships of Grade-7 Parental Support for College Education, Family SES, and Educational Expectations of Youths in prediction of Educational Achievement of Youths in Adulthood (Model 2).

Predictor	Mediator(s)	Outcome	β_ind_	SE	Z-Value
Parental Support for College Education	Academic Commitment of Youths	Educational Achievement	0.003	0.001	1.167 *
	Educational performance of Youths	Educational Achievement	0.041	0.008	5.164 ***
	Academic Commitment of Youths and Educational performance	Educational Achievement	0.005	0.008	5.164 **
Family SES	Academic Commitment of Youths	Educational Achievement	0.000	0.000	−0.028
	Educational performance of Youths	Educational Achievement	0.003	0.001	4.284 ***
	Academic Commitment of Youths and Educational performance	Educational Achievement	0.000	0.000	−0.028
Educational Expectation of Youths	Academic Commitment of Youths	Educational Achievement	0.008	0.004	2.223 *
	Educational performance of Youths	Educational Achievement	0.076	0.010	7.936 ***
	Academic Commitment of Youths and Educational performance	Educational Achievement	0.014	0.002	6.069 ***

Note. Family SES = Family Socioeconomic Status, ** p* < 0.05; ** *p* < 0.01; *** *p* < 0.001.

**Table 5 ijerph-20-03279-t005:** Moderated Effects of Parental Support for College Education by Family SES and Educational Expectation of Youths by Family SES in Grade 7 on Youths’ Grade-8 Academic Commitment, Grade-9 Educational Performance, and Educational Achievement in Adulthood (Model 2).

Predictor	Outcome	β_int_	SE	Z-Value
Parental Support for College Education X Family SES	Academic Commitment of Youths	0.000	0.001	−0.090
	Educational Performance of Youths	0.002	0.001	2.060 *
	Educational Achievement of Youths	0.002	0.001	1.499
Educational Expectation of Youths X Family SES	Academic Commitment of Youths	0.000	0.001	0.358
	Educational Performance of Youths	0.003	0.001	2.925 **
	Educational Achievement of Youths	0.002	0.001	2.917 **

Note. Family SES = Family Socioeconomic Status, * *p* < 0.05; ** *p* < 0.01.

## Data Availability

The data of LSAY are available at The Inter-university Consortium for Political and Social Research (ICPSR), researchers can visit https://www.icpsr.umich.edu/web/ICPSR/studies/30263 (accessed on 30 October 2022).

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
