# Peer review of "Family and Individual Contexts of Middle-School Years and Educational Achievement of Youths in Middle-Aged Adulthood"

_ijerph, 2023, doi:10.3390/ijerph20043279_

Round 1

Reviewer 1 Report

Dear authors,

Thank you for this interesting and relevant study. I really enjoy to read it. I only suggest more detail when implications for policy and practice are referred to. Maybe, some specific suggestions can be provided. Also, check minor errors related to text editing.

Best

Author Response

Thank for your reviewing the manuscript titled “Family and Individual Contexts of Middle-School Years and Educational Achievement of Youths in Middle-Aged Adulthood”, and our responses to the reviewers’ comments are below:

For reviewer 1

1) Thank you for this interesting and relevant study. I really enjoy to read it. I only suggest more detail when implications for policy and practice are referred to. Maybe, some specific suggestions can be provided. Also, check minor errors related to text editing.

Reply:  Now the whole manuscript has been revised to make the study relationships between youths’ grade-7 parental support for college education, family SES, and educational expectations in contribution to their educational achievement in adulthood of mid-thirties through their development of grade-8 academic commitment and grade-9 educational performance more accurate and coherent, in which more elaborations about how the findings of the current study correspond with the theories used to construct the study relationship are presented in the section of “Discussion” that are followed by the suggestions of relevant and possible policy and practice.

Reviewer 2 Report

I think this manuscript provided information that is valuable to the readers and has a lot of strong support for the hypotheses. The suggestions I have are to improve the readability of the paper. The first two sentences of the abstract are very long and run on. I suggest breaking them up to make them clearer. They are confusing. 

Lines 51-55 are also run on sentences. Please split into two to make it more clear.

The facmly context and educational development of youths section is long and seems to repeat. I suggest cutting it and combining parts. I feel like there are a lot of repeats. Line 109 is unclear.

There are three or four different theories introduced in the background section, yet none of the results come back to talk about them in the discussion. I believe these should be limited to one or two theories and then drawn back to in the discussion. Indicate whether the results support or refute the theory/ies and why.

Lines 201-221 do not seem to fit in this section. THey are more family based.

The results seem straightforward and strongly support the hypotheses. The discussion seems appropriate based on the results.

Author Response

Thank for your reviewing the manuscript titled “Family and Individual Contexts of Middle-School Years and Educational Achievement of Youths in Middle-Aged Adulthood”, and our responses to the reviewers’ comments are below:

For reviewer 2

1) I think this manuscript provided information that is valuable to the readers and has a lot of strong support for the hypotheses. The suggestions I have are to improve the readability of the paper. The first two sentences of the abstract are very long and run on. I suggest breaking them up to make them clearer. They are confusing. 

Reply: The first two sentences of the abstract are now revised to avoid run-on presentations, and the whole manuscript has been proofread again to make the presentations more concise and accurate.

2) Lines 51-55 are also run on sentences. Please split into two to make it more clear.

Reply : Lines 51-55 are now revised to make the presentations more accurate and coherent.

3) The facmly context and educational development of youths section is long and seems to repeat. I suggest cutting it and combining parts. I feel like there are a lot of repeats. Line 109 is unclear.

Reply: the presentations written in the section “Family Context and Educational Development of Youths” are not long-winded and surplus, in which as the manuscript needs to construct the theoretical relationships of how grade-7 parental support for college education and family SES, contributing to youths’ educational achievement in adulthood through their development of grade-8 academic commitment and grade-9 educational performance, hence logic and systematic arguments must be needed. If readers read carefully, they will find the structured study relationships are coherently structured. For line 109, it is now rewritten to be clearer.

4)There are three or four different theories introduced in the background section, yet none of the results come back to talk about them in the discussion. I believe these should be limited to one or two theories and then drawn back to in the discussion. Indicate whether the results support or refute the theory/ies and why.

Reply: Now the life course perspective, intergenerational transmission perspective, social capital theory, family stress theory, self-determination theory, and expectancy value theory that are used to construct the theoretical study relationships are elaborated in the section of “Discussion” to state how the findings support the arguments to let readers take a more integrative view on how these theories can validly buttress the study relationships and support its hypotheses.

5) Lines 201-221 do not seem to fit in this section. THey are more family based.

The results seem straightforward and strongly support the hypotheses. The discussion seems appropriate based on the results.

Reply: Lines 201-211(I guess the line numbers are the sentences that the reviewer would like to concern) are used to introduce relevant research for how the effects of Family SES on educational development of youths would be moderated by parental support for college education and educational expectations of youths to be a less degree (although the findings turn out the opposite). Hence, they are needed in the presentations.